# Using Small Non-Coding RNAs in Extracellular Vesicles of Semen as Biomarkers of Male Reproductive System Health: Opportunities and Challenges

**DOI:** 10.3390/ijms24065447

**Published:** 2023-03-13

**Authors:** Sara Larriba, Francesc Vigués, Lluís Bassas

**Affiliations:** 1Human Molecular Genetics Group, Bellvitge Biomedical Research Institute (IDIBELL), Hospitalet de Llobregat, 08908 Barcelona, Spain; 2Urology Service, Bellvitge University Hospital, ICS, Hospitalet de Llobregat, 08908 Barcelona, Spain; 3Laboratory of Andrology and Sperm Bank, Andrology Service-Fundació Puigvert, 08025 Barcelona, Spain

**Keywords:** urogenital diseases, male infertility, prostate cancer, semen, extracellular vesicles, sncRNAs, non-invasive biomarker, diagnosis

## Abstract

Reproductive dysfunction and urogenital malignancies represent a serious health concern in men. This is in part as a result of the absence of reliable non-invasive tests of diagnosis/prognosis. Optimizing diagnosis and predicting the patient’s prognosis will affect the choice of the most appropriate treatment and therefore increase the chances of success and the result of therapy, that is, it will lead to a more personalized treatment of the patient. This review aims firstly to critically summarize the current knowledge of the reproductive roles played by extracellular vesicle small RNA components, which are typically altered in diseases affecting the male reproductive tract. Secondly, it aims to describe the use of semen extracellular vesicles as a non-invasive source of sncRNA-based biomarkers for urogenital diseases.

## 1. Introduction

Reproductive dysfunction and urogenital malignancies in males affect the health and quality of life of many men. Of all these pathologies, prostate cancer is diagnosed in one of every six individuals [1], and is the second most frequent cancer in men. In addition, between 5 and 7% of men in Western countries present infertility or subfertility [2]. Both prostate cancer and fertility problems affect organs of the male reproductive system, all of which contribute to different extents to the production of semen. These diseases represent a serious health concern in men. This is in part as a result of the absence of reliable non-invasive tests of diagnosis/prognosis. Thus, in most cases, diagnosis/prognosis prediction is based on tissue biopsy analyses, which, due to tissue heterogeneity and the pitfalls of the sampling procedures, are often not informative. Optimizing diagnosis and predicting the patient’s prognosis will affect the choice of the most appropriate treatment and therefore increase the chances of success and the result of therapy, that is, it will lead to a more personalized treatment of the patient. For these reasons, the identification of precise and preferably non-invasive diagnostic/prognostic biomarkers for urogenital diseases is essential.

In general, both healthy and pathological cells release various types of membranous structures that are called small extracellular vesicles (sEVs) into the fluids. These mediate paracrine signaling and participate in intercellular communication [3,4], both at short and long distances (through the bloodstream, lymph, or other fluids), with important implications for biology, disease, and medicine [5]. The term sEV refers to an EV smaller than 200 nm in diameter, which includes both microvesicles, released from the plasma membrane, and exosomes, generated from endosomal multivesicular bodies. These vesicles participate in cellular regulation by fusing with the cytoplasmic membrane of the target cell and discharging the cargo they transport. sEVs are associated with apoptosis, coagulation, and inflammation, and can control the polarity of cell development and differentiation. They may also be involved in the growth, survival, differentiation, transmission, and stress reactions of cells [6]. Importantly, tumor cells can discharge sEVs, which are involved in tumor growth, metastasis, and resistance to chemotherapy [7].

sEVs contain molecules from their cell of origin, and so can inform on its characteristics and cellular health. Due to this, in recent years there has been a growing interest in the study of sEVs in tissue and fluids in order to determine their involvement in pathology and to evaluate their use as a source of biomarkers.

## 2. Semen Extracellular Vesicles

It is important to note that semen contains one of the highest reported concentrations of extracellular vesicles of any body fluid. These sEVs originate from the different organs of the male reproductive system: prostate (these sEVs, also known as prostasomes, represent 40% of semen sEVs), epididymis (epididysomes), seminal vesicles, and testicles [8].

Although it is developmentally mature, sperm exiting the seminiferous tubule still requires further modification. Changes in sperm morphology and post-testicular sperm function rely on sperm interaction with the intraluminal fluid during transit through the epididymis and vas deferens, suggesting that EVs in male reproductive biofluids may participate in intercellular communication after spermatogenesis. Additionally, in the female reproductive system, sperm acquisition of various functions including increased motility, transfer of cargoes, and ability to undertake the acrosome reaction is mediated through the interaction between the sperm and semen EVs derived from the organs of the male reproductive tract. Thus, semen EVs have a relevant physiological function related to the process of fertility [8] (point 1), but they are also involved in pathological states of the male reproductive tract such as the development and dissemination of prostate cancer [9,10] (point 2).

### 2.1. Semen sEVs and Fertility

Seminal plasma (SP) sEVs have been proposed as a means of selectively transporting and delivering various regulatory molecules to the female reproductive system to facilitate conception and contribute to fertilization [8] and embryo implantation [11], the latter by targeting endometrial stromal cells. Strikingly, as a first step, semen sEVs transfer their content directly to the male germ cell in the acid medium of the vagina [12] to protect sperm and modulate their activity. The numerous enzyme systems, small signaling molecules, and neuroendocrine markers associated with semen sEVs suggest that these vesicles directly or indirectly play a complex role in the regulation of sperm viability and function [13]; (reviewed in [14,15]).

(A)Direct functions
-Motility of sperm:Sperm motility is vital for natural fertility and thus survival after ejaculation into the female reproductive tract [16]. Semen EVs improve the progressive motility of sperm [17] in the acid medium of the vagina [18,19] by increasing intracellular calcium.-Sperm capacitation and acrosome reaction:Both sperm capacitation and the reaction of the acrosome are crucial steps for the sperm to acquire the ability to fertilize the oocyte, by allowing it to cross the zona pellucida of the oocyte to achieve fertilization [20]. Semen sEVs regulate these processes and prevent them from occurring prematurely after fusion between EVs and sperm [21]. On the other hand, this fusion also provides essential hydrolases for the acrosome reaction [22], as well as making the sperm more sensitive to progesterone, one of the stimulators of the acrosome reaction [23,24].(B)Indirect functionsSemen sEVs help to protect sperm after ejaculation from the immunological point of view, as they provide CD46 and CD59 proteins that help the sperm regulate complement activity of the female reproductive tract [25]; they also present antioxidant attributes that help avoid oxidative stress of sperm [26], antibacterial properties [27], and coagulant activity [28].

### 2.2. Semen sEVs and Prostate Cancer

Tumor-derived sEVs have been described as being involved in the development and dissemination of PCa [9,10]. In the tumor environment, EVs derived from malignant cells carry genetic and proteinic messages to the target cells which reduce their immune response. In this way, they influence the homeostasis of the surrounding environment and the progression of the tumor in the donor tissue. Reproductive system sEVs are directly secreted to semen and some of them can also reach blood and plasma. In this way, through the bloodstream, they contribute to the formation of the premetastatic niche in other body tissues (metastasis). The EVs secreted by cancer cells into the blood have a cancer-specific molecule content [29] such as a high proportion of oncoproteins and RNAs [30]. These EVs not only intervene in tumor progression, but also in the communication between tumors and the immune system, contributing to the drug-resistant character of cancer cells.

## 3. sncRNA EV Content as a Biomarker

sEVs are characterized by a high content of cholesterol and sphingomyelin, as well as by a very complex protein composition. EVs also carry RNA that can be transferred to other cells, modulating the function of the recipient cells [3,4]. This RNA can be found in the form of mRNAs and non-coding regulatory RNAs (including small and long ncRNAs) [3] (Figure 1).

sEVs in semen contain a very important population of small ncRNAs (sncRNA, 20 to 100 nucleotides), which have a significant impact on the regulation of gene expression through a variety of epigenetic and post-transcriptional mechanisms [31]. RNA biotypes identified within sncRNAs include microRNAs (miRNAs), PIWI-interacting RNAs (piRNAs), and endogenous interfering RNAs (endo-siRNAs), which are critical regulators of germ cell development. There are also other more recently described molecules such as transfer RNA (tRNA)-derived small RNAs (tsRNAs) and ribosomal RNA (rRNA)-derived small RNAs (rsRNAs), whose role in reproduction and fertility is yet to be fully settled (reviewed in [32]). MiRNAs (21%) and tsRNAs (16%) represent the most abundant sEV sncRNAs in semen [8] (Figure 2A). The sncRNA profile of sEVs in semen is unique and differs from the profile found in sEVs from other fluids [8].

Since sncRNAs are encapsulated inside sEVs, they remain stable and protected from RNAse. Due to these attributes, sEV sncRNAs are considered relevant for study as reliable biomarkers. Additionally, EVs contain molecules of the progenitor cell, so these EVs in the fluids can reflect the identity, characteristics, and health of the cell or tissue of origin. This is highly relevant in the context of the study of the content of sEVs as biomarkers. Specifically, in recent years, an increasing number of studies have been published that evaluate the miRNAs of sEVs in fluids as diagnostic biomarkers for physiological processes, such as the immune response, and pathological processes such as cancer, with promising results. The miRNAs participate in the post-transcriptional regulation of the expression of the genes: the specific binding of these small molecules of RNA to the transcripts affects their stability and activates transcript degradation. The tissue-specific expression profile and the high stability of these molecules mean that miRNAs, and other regulatory small RNAs, are becoming more and more relevant as biomarkers for the diagnosis or prognosis of different pathological conditions.

In recent years, it has been shown that ncRNAs play an important role in tumor progression and metastasis by their dissemination through fluids to regulate cancer cells and/or cancer stem cells, which suggests that extracellular ncRNAs in circulating sEVs can be used not only for diagnosis but also as prognostic biomarkers [33].

## 4. The Small RNA Population Signature Is Different between the Organs of the Reproductive Tract

As previously mentioned, semen contains a heterogeneous composition of sEVs, which are released from the different organs of the male reproductive tract (Figure 2A), and these sEVs contain molecules from their cell of origin. Levels of these molecules can reflect the features and the health of that original cell. Certain sncRNA subtypes, such as miRNAs, have been associated with gene deregulation in diseases affecting organs of the male reproductive tract, including prostate cancer [34] and spermatogenic disorders [35]. These attributes support the idea that the study of the small RNA content of semen sEVs can reflect the pathophysiological state of male reproductive organs (Figure 2B). The small RNA population varies between tissues; therefore, it is essential to know the expression pattern of the small RNAs in each of the organs of the male reproductive tract to be able to correctly interpret semen sEV sncRNA expression behavior and use it as a potential biomarker of urogenital disease.

### 4.1. Profile of Small RNA Populations in Human Testis

Spermatogenesis is a highly orchestrated developmental process which occurs in the testicular seminiferous tubules, and by which primordial germ-cells or spermatogonia develop into haploid spermatozoa. Production of sperm depends on precise developmental stage- and germ-cell type-specific gene expression. Thus, spermatogenesis is heavily dependent on post-transcriptional regulatory processes, and miRNAs have emerged as important regulators of these events [36,37,38]; miRNAs specifically expressed in the germline, such as miR-122, miR-449a, and miR-34b/c, are especially relevant [35,39,40]. In human testes, it is not only miRNAs that are highly abundant but also piRNAs, and tsRNAs at a lesser extent [41]. Interestingly, in physiological conditions, piRNAs are almost exclusively found in the male gonad, mainly in the germline [42,43,44].

### 4.2. Epididymis

The epididymis is a convoluted tube at the posterior part of the testis, where the sperm mature and are stored. When the sperm leaves the testis, it harbors a specific profile of small RNA, including all major RNA classes, with piRNAs being the most abundant sncRNAs. During its post-testicular maturation, the spermatozoon does not generate sncRNAs, but acquires them exogenously thanks to its fusion with extracellular vesicles [45,46,47,48]. In the epididymal lumen, epididymal epithelial cells secrete epididymosomes that contain small RNAs and proteins. Epididymosomes can deliver miRNAs, tsRNAs, and other sncRNAs to sperm that are transported from the proximal to distal epididymal region (from the caput to the cauda), resulting in changes in sncRNA patterns in mature sperm, with a notable enrichment of tRNA fragments (tRFs) [47] and other miRNAs [49] and a global loss of piRNAs.

Functional studies of epididymal sncRNAs have revealed a relevant role in epididymal region-specific gene regulation and epithelium–sperm interactions. Analysis of epididymosomes revealed that their cargo contains most small RNA species, including miRNAs, tsRNAs, and RNAs derived from snRNAs, snoRNAs, and rRNAs [50]. Some semen EVs contained miRNAs such as the miR-888 family (located in epididymis-enriched cluster of chromosome X), which have been described as being specifically enriched in the distal region of epididymis [39,40,51].

### 4.3. Prostate

Expression profiles of miRNAs obtained from malignant and benign prostate tissues differ significantly [52,53,54]. In recent years, PCa-related miRNAs have been identified and proposed as a tool for early and specific detection of the disease. Recent studies have revealed that abnormal expression of a large number of tsRNAs also occurs in PCa [55,56], contributing to DNA synthesis, cell viability, and cell proliferation.

Approximately 40% of semen is derived from prostatic tissue. At ejaculation, cauda epididymal spermatozoa are mixed with secretions from the reproductive tract accessory glands such as the prostate. The use of semen, rather than other fluids such as urine after prostate massage, is preferable as a non-invasive source of information of prostate health because semen represents a liquid biopsy from the whole prostatic gland whereas a sample is only obtained from the posterior part of the gland when urine is used [57].

## 5. Scenarios of Urogenital Pathology for the Study of the Content of Extracellular Vesicle sncRNAs in Semen

Exploring non-invasive sources of molecular markers for male infertility and reproductive cancer will lead to the elaboration of more rational clinical procedures based on genetic information. It is relevant to identify useful non-invasive biomarkers to be able to provide a more personalized medicine with the aim of improving screening, early detection, and disease follow-up. Here, we summarize the results obtained at present based on semen sEV-sncRNAs (Figure 2C).

### 5.1. Male Infertility Due to Poor Semen Quality

Male infertility, in a high proportion of cases, is characterized by a decreased semen quality in terms of number, progressive motility, and morphology of spermatozoa [58]. Some studies have shown that altered testicular miRNA expression accompanies a low number of sperm in semen such as oligoasthenozoospermia [59] and non-obstructive/secretory azoospermia [60], mostly associated with changes in the cell type composition (lack of germ cells in varying degrees) in pathological testis. A signature of differentially expressed miRNAs at tissular but also at cellular level in infertile patients with severe disruption of spermatogenesis has also been described by our group [35]; decreased cellular miRNA content in developing germ cells of spermatogenic disorders also suggests that the cellular miRNA content of mature germ cells depends heavily on the efficacy of the spermatogenic process.

Interestingly, miRNA dysregulation was also observed in semen extracellular vesicles from oligoasthenozoospermic individuals [59,61]. The authors suggest the use of these miRNAs to understand the molecular mechanisms underlying male infertility in a non-invasive way.

Regarding azoospermia, which is characterized by the absence of sperm in ejaculate, it is a relatively common form of male infertility that occurs as a result of a failure in testicular spermatogenesis (secretory azoospermia with no or few sperm in the testicle) or an obstruction in the genital tract (obstructive azoospermia with preserved spermatogenesis). Being able to discriminate cases of obstructive azoospermia is of great relevance in the clinical setting, since these individuals have a good chance of obtaining sperm from testicular biopsy for assisted reproductive treatment. Recently, our group demonstrated the usefulness of the levels of miRNAs contained in semen sEVs as a biomarker for the presence of testicular sperm in azoospermic individuals [40]. The aim of our study was to profile the expression levels of miRNA contained in the semen exosomes to evaluate their potential as non-invasive biomarkers that can contribute to the diagnosis of azoospermia. Among a total of 623 miRNAs tested in the study, we described a signature of differentially expressed miRNAs in semen with no sperm and proposed the expression values of sEV miR-31-5p from semen as a predictor for the origin of azoospermia with high sensitivity and specificity. Additionally, a model including the expression values of two miRNAs was also described as useful for predicting the presence of residual sperm in the testis in severe spermatogenic disorders; this would help to avoid unnecessary testicular biopsies for reproductive treatment. Analysis in silico of selected miRNA target genes suggested a functional pathophysiological relevance of miRNA–target gene interaction in spermatogenic disorders.

Some additional efforts have been made to identify sEV-contained sncRNAs to discriminate the presence or the absence of sperm in a testicular biopsy with a spermatogenic disorder with high diagnostic accuracy. Outcome prediction of TESE (testicular sperm extraction) [40] and micro_TESE (microdissection TESE) [62,63] is relevant before azoospermic individuals are referred for assisted reproduction treatment (ART) because they can only benefit from ART procedures if testicular sperm is present.

Additionally, on the topic of male infertility related to deficient sperm motility, men with less than 32% of progressively motile spermatozoa in the ejaculate are classified as having asthenozoospermia. Hong and collaborators [64] demonstrated that semen sEVs of asthenozoospermic patients have decreased levels of piRNAs compared to normozoospermic individuals, probably due to a loss of function of the MitoPLD protein, which is involved in piRNA biogenesis.

### 5.2. Unexplained Male Infertility

In a high percentage of infertile men, their etiology is closely related to differences in semen quality. However, in a significant proportion of infertile individuals the systematic and complete diagnostic evaluation is absolutely normal and the etiology of infertility remains unknown. These males can present functional defects in sperm only detectable during the performance of assisted reproduction techniques.

Many efforts have been made to develop new diagnostic tests on cellular fractions of semen that will provide accurate information on the fertilizing capacity of human sperm [65] and predict pregnancy after reproductive treatment [66], but still none of them meet the standards sufficiently to be applicable for clinical purposes.

As previously mentioned, sEVs in semen regulate sperm capacitation and acrosome reaction in the female reproductive tract, both of which are necessary for the sperm to acquire the ability to fertilize the oocyte. Specifically, semen extracellular vesicle-contained miRNAs have been demonstrated to be involved in sperm capacitation (miR-21-5p) [67] and sperm apoptosis (miR-222) [68]. Accordingly, the study of ncRNA content of semen exosomes emerges as an opportunity to determine their involvement in severe defects of sperm function (before and after conception) in normozoospermic infertile individuals and explain their decreased fertility rates.

The identification of molecular biomarkers of fecundity in EVs in semen would additionally give information about the origin of certain male deficiencies that affect fertility. This, in turn, will provide prognostic information on the possibilities of pregnancy after the assisted reproduction techniques as well as helping physicians to decide on the most suitable reproductive treatment for the couple.

### 5.3. Prostate Cancer

Although prostate cancer (PCa) affects the health and quality of life of many men, there are still no accurate non-invasive methods of diagnosis for this pathology. Despite the widespread use of the prostate specific antigen (PSA) in serum as a diagnostic test, a large number of false positives and false negatives are being reported; this is because, despite PSA having been recognized as specific for prostatic tissue, it has low specificity for malignant prostate disease (PSA also increases in non-malignant conditions such as benign prostatic hyperplasia and prostatitis). Specifically, attention should be paid to patients with PSA levels of 4 to 10 ng/mL, for whom the detection rate of PCa is merely 20%. For this reason, confirmation of the diagnosis is usually needed, which is based on the practice of tissular biopsies. Consequently, new and more effective non-invasive biomarkers for PCa are needed that can help better identify which patients should undergo confirmatory diagnostic tests in order to avoid a widespread use of invasive methods and unnecessary biopsies. Numerous miRNAs in prostatic tissue have been found to be deregulated when associated with the development and/or progression of PCa [53].

Similarly, in the same way as occurs in healthy cells, prostate epithelial cancer cells produce extracellular vesicles (prostasomes). Quantitative and qualitative changes in microRNA composition of semen sEV isolated from prostate cancer patients have been reported [69]. Profiling the expression level of miRNAs contained in the semen exosomes enabled the assessment of their usefulness as non-invasive biomarkers for the diagnosis of PCa. Altered miRNA expression profile in seminal plasma from PCa men with moderately elevated PSA levels (4–15 ng/mL) were correlated with established risk factors. A model including the expression values of several miRNAs, in particular miR-142-3p, miR-142-5p, and miR-223-3p, was described as useful for predicting the presence of cancer cells in the prostate as well as for discriminating prostate cancer from other benign conditions that affect prostate (benign prostatic hyperplasia)**,** suggesting it could be helpful as a molecular test that could complement PSA analysis as a PCa diagnosis biomarker [69].

In this context, there is an urgent need for new and more effective non-invasive biomarkers with a high prognostic capacity for assessing the risk of growth and spread of PCa and/or response to treatment. These biomarkers would be of special value for patients with distinct outcomes although they have apparently similar clinical characteristics, and would improve prediction and the early detection of disease evolution as well as treatment response, which would have further consequences such as helping health care professionals make better therapeutic decisions.

**Figure 2 ijms-24-05447-f002:**
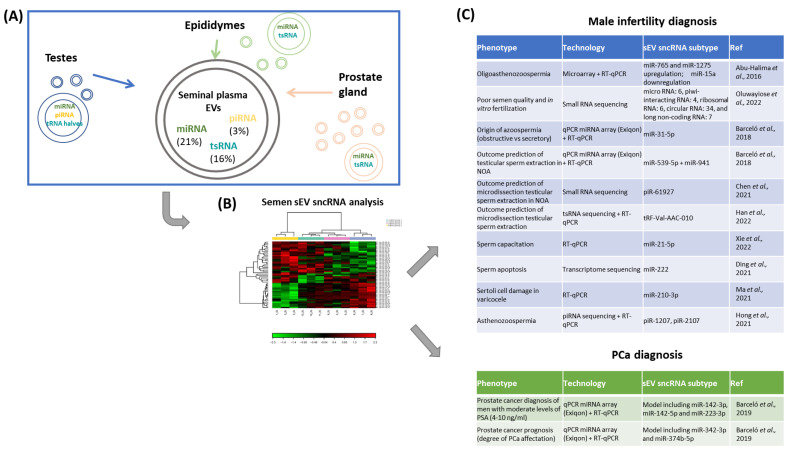
Uncovering semen EV sncRNAs as biomarkers for urogenital diseases. (**A**) Semen contains a heterogeneous composition of sEVs, released from the different organs of the male reproductive tract. This fact determines the proportion of the different sncRNA subtypes identified in semen sEVs as the profile of small RNA population varies between tissues in the male reproductive tract. (**B**) This attribute should be considered for a proper interpretation of the sncRNA expression analysis (a cluster analysis is shown as an example) and the selection of semen EV sncRNA as biomarkers for urogenital disease. (**C**) Male reproductive seminal sEV sncRNA of interest are summarized in the tables [39,58,60,61,62,63,66,67,68].

## 6. Technical Issues to Consider for a Proper Interpretation of Results on Semen sEVs

### 6.1. Impact of Surgical Procedures for Contraception

The number of couples selecting male vasectomy as a contraceptive method has been increasing in recent years. The practice of vasectomy affects the concentration of certain semen sEV sncRNAs because the fluid from the testis and epididymis cannot reach semen, so whether the subject has undergone vasectomy should be taken into account for the accurate interpretation of results on semen for biomarker discovery/validation in urogenital diseases [69]. 

### 6.2. Impact of Extracellular Vesicle Isolation Methods and Preclinical Variables on Downstream sncRNA Analysis in Semen

Several exosome-EV isolation technologies have been optimized for their use in semen, and their EV purifying effectiveness (EV quantity, size, and transmembrane protein composition, as well as the quality of RNA they contain) and the impact on the down-stream analysis of miRNAs, compared against the standard use of ultracentrifugation, have been further evaluated [70]. Our study provides evidence that the exosome-EV isolation method has a great impact on the analysis of the miRNAs they contain; it will be important to take this into consideration when implementing the use of semen exosomal/EV miRNAs as a diagnostic tool in the clinical laboratory. The observed over-representation of certain miRNAs from vesicles obtained by a particular isolation method suggests that some miRNAs are likely to be associated with specific vesicle sets enriched by this particular method of isolation. Therefore, the method used for EV isolation can produce variations in EV concentration as well as determine the composition of sEV subpopulations (such as microvesicles and exosomes) in nanovesicle preparation, introducing a bias for subsequent miRNA analysis. Accordingly, until clear markers for delineation between microvesicles and exosomes are established, the results of the analysis of miRNAs contained in EVs cannot be directly extrapolated between different EV isolation methods for clinical application due to the possibility of obtaining misleading results and conclusions.

### 6.3. Impact of Profiling Methods for Biomarker Testing

Due to the small size of sncRNAs and their low abundance in human fluids, sncRNA expression profiling is technically challenging. Since the discovery of sncRNA, many platforms have been developed for their expression quantification, specially tested for miRNAs, such as small RNA sequencing, microarray hybridization, and reverse transcription-quantitative PCR (RT-qPCR). Several studies in human biofluids, such as serum and plasma, have assessed platform performance in terms of reproductivity, sensitivity, accuracy, and specificity, and indicated that each method has its strengths and weaknesses, which can impact on differential expression analysis [71,72]. Other factors such as normalization and data imputation methods are of relevance to be considered, and will help for a standardized quantification of semen sEV sncRNAs for their clinical use as biomarkers for urogenital diseases.

## 7. Future Perspectives of Study

It must be kept in mind that there is limited information regarding the mechanisms that regulate sncRNA levels and/or function in physiological and/or pathological context. Any alteration of these regulatory mechanisms could explain potential discrepancies in the published results on the usefulness of sncRNA as biomarker.

In this context, recent publications have contributed with relevant results. Several studies have suggested that certain ncRNAs contain multiple binding sites for miRNAs, and act as “sponges” of miRNAs, regulating their concentration, function, and/or activity. Some examples of these include the long non-coding RNAs (lncRNAs) and the circular exonic RNAs (circRNAs).

The lncRNAs, among which the intergenic lncRNAs (lincRNAs) stand out [73,74], perform various functions in a wide variety of important biological processes. Recent studies have shown that prostate cancer EVs present increased content of lncRNAs enriched in miRNA binding regions [75]. These lncRNAs also have binding motifs for RBP (RNA binding proteins). A semen EV lncRNA panel has been suggested as being useful for cases of secretory azoospermia to predict the presence of testicular spermatozoa [76].

The circRNAs are also capable of sequestering and binding mature miRNAs [77]. It is interesting to note that these circRNAs have also been identified in sEVs at higher concentration than in the cells of origin [78].

The identification of a lncRNA/circRNA–mRNA–sncRNA regulation network in semen sEVs from individuals with urogenital diseases could provide useful information about the molecular mechanisms involved in each of the diseases, and reflect an important part of the molecular events in the male genital tract.

## 8. Conclusions

The analysis of the content of semen sEVs, and particularly of sncRNAs, can reflect relevant molecular events in the male genital tract associated with the characteristics of disease. The findings described in this review contribute to the search for the most valuable genetic markers that are potentially useful as tools for the appropriate management of urogenital disease diagnosis and/or prognosis in a non-invasive way. Analysis of semen sEV sncRNAs represents a promising and reliable option either as a stand-alone tool, or for use in addition to traditional biomarkers, in order to provide clinically useful information of disease attributes without the use of unnecessary biopsies.

## Figures and Tables

**Figure 1 ijms-24-05447-f001:**
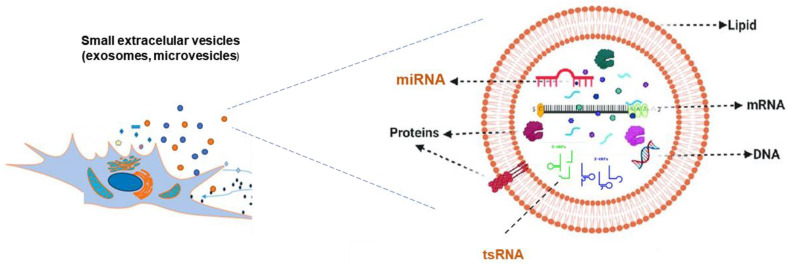
Both healthy and pathological cells release various types of membranous structures that are called small extracellular vesicles (sEVs; <200 nm of diameter) into the fluids. The sEVs are characterized by a high content of cholesterol and sphingomyelin. sEVs carry a very complex molecular load including proteins, DNA and RNA; RNA can be found in the form of mRNAs and non-coding regulatory RNAs, including small (depicted in orange) and long ncRNAs. All of this content can be transferred to other cells participating in paracrine signaling.

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
