# Peer review of "Using Small Non-Coding RNAs in Extracellular Vesicles of Semen as Biomarkers of Male Reproductive System Health: Opportunities and Challenges"

_ijms, 2023, doi:10.3390/ijms24065447_

Round 1

Reviewer 1 Report

The manuscript presents a review about the potential use of small non-coding RNA contained in seminal extracellular vesicles as diagnostic markers of several male reproductive pathologies. As male infertility is an emerging problem in a modern society, this subject is worth reviewing.  

The manuscript is well structured and presents comprehensive review of literature, with majority of referred articles being published in past few years. All important aspects were properly discussed. It is enriched in several figures, including a table with scnRNA that can be potential markers of reproductive pathologies in men.  

I have only minor remarks and besides them, I find the manuscript valuable to be published.

Minor remarks:

- The title could be shorter and more ‘catchy’.

- End of introduction: ‘Importantly, tumour cells can discharge sEVs, which are involved in tumour growth, metastasis, and resistance to chemotherapy’ – reference to this statement can be added.

- On page 4 one sentence is a different font

- Page 8, in part 5.3. Prostate cancer: ‘Vasectomy can affect semen miRNA concentration, and thus, was considered for the correct interpretation of results.‘ – vasectomy should not affect prostasomes, right? And the influence of vasectomy is mentioned in the next part, I think it should be removed from part 5.3.  

- I really liked that the Authors included a paragraph about technical issues (part 6), other limitation (e.g. the need of specialized laboratory, high price, the need for standardization) can be mentioned.

Author Response

Reviewer 1

Comments and Suggestions for Authors

The manuscript presents a review about the potential use of small non-coding RNA contained in seminal extracellular vesicles as diagnostic markers of several male reproductive pathologies. As male infertility is an emerging problem in a modern society, this subject is worth reviewing. 

The manuscript is well structured and presents comprehensive review of literature, with majority of referred articles being published in past few years. All important aspects were properly discussed. It is enriched in several figures, including a table with scnRNA that can be potential markers of reproductive pathologies in men. 

I have only minor remarks and besides them, I find the manuscript valuable to be published.

Answer: We thank the reviewer for his/her positive valuation of the work.

Minor remarks:

- The title could be shorter and more ‘catchy’.

Answer: Title has been shortened as suggested “SMALL NON-CODING RNAS IN EXTRACELLULAR VESICLES OF SEMEN AS BIOMARKERS OF MALE REPRODUCTIVE SYSTEM HEALTH: OPPORTUNITIES AND CHALLENGES”

- End of introduction: ‘Importantly, tumour cells can discharge sEVs, which are involved in tumour growth, metastasis, and resistance to chemotherapy’ – reference to this statement can be added.

Answer: References have been included as suggested.

- On page 4 one sentence is a different font

Answer: I have checked that the whole manuscript is written with the same text font

- Page 8, in part 5.3. Prostate cancer: ‘Vasectomy can affect semen miRNA concentration, and thus, was considered for the correct interpretation of results.‘ – vasectomy should not affect prostasomes, right? And the influence of vasectomy is mentioned in the next part, I think it should be removed from part 5.3

Answer: Vasectomy will affect the proportion of semen vesicles that come from testis and/or epididymis. This issue not only affect spermatogenic disorder studies. In the case of prostate cancer, the practice of vasectomy has also to be taken into account when selecting patients to be included into the study. If one group is enriched in vasectomized patients, it can result into false positive/negative results.

We have removed the sentence in part 5.3. as referee suggested and have modified the paragraph in part 6.1.

- I really liked that the Authors included a paragraph about technical issues (part 6), other limitation (e.g. the need of specialized laboratory, high price, the need for standardization) can be mentioned.

Answer: We thank the reviewer for his/her suggestion and have included an extra paragraph in section 6, referring the impact of profiling methods for biomarker testing (section 6.3.)

Reviewer 2 Report

The article (review) was nicely written and arranged to show information about extracellular vesicles in sperm as indicators for certain diseases or other male dysfunction.

The paper discussed the usage of biomarkers found in extracellular vesicles, their effect on sperm performance, and their putative relationship to male illness. Some extracellular vesicles, such as miRNA, tRNA, and others, have been characterized as being available in the sperm and maybe as a biomarker for specific purposes.

The ability to use extracellular vesicles as a noninvasive approach is a fantastic opportunity. Still, there is a long road ahead to validate the exact role of each biomarker.

Author Response

Comments and Suggestions for Authors

The article (review) was nicely written and arranged to show information about extracellular vesicles in sperm as indicators for certain diseases or other male dysfunction.

The paper discussed the usage of biomarkers found in extracellular vesicles, their effect on sperm performance, and their putative relationship to male illness. Some extracellular vesicles, such as miRNA, tRNA, and others, have been characterized as being available in the sperm and maybe as a biomarker for specific purposes.

The ability to use extracellular vesicles as a noninvasive approach is a fantastic opportunity. Still, there is a long road ahead to validate the exact role of each biomarker.

Answer: We thank the reviewer for his/her positive valuation of the work.